# Efficient $k$-NN Search with Cross-Encoders using Adaptive Multi-Round CUR Decomposition

**Nishant Yadav[†], Nicholas Monath[◇], Manzil Zaheer[◇], and Andrew McCallum[†]**
[†] University of Massachusetts Amherst   [◇]Google Research
{nishantyadav, mccallum}@cs.umass.edu   {nmonath,manzilzaheer}@google.com

## Abstract

Cross-encoder models, which jointly encode and score a query-item pair, are prohibitively expensive for direct $k$-nearest neighbor ($k$-NN) search. Consequently, $k$-NN search typically employs a fast approximate retrieval (e.g. using BM25 or dual-encoder vectors), followed by reranking with a cross-encoder; however, the retrieval approximation often has detrimental recall regret. This problem is tackled by AN-NCUR (Yadav et al., 2022), a recent work that employs a cross-encoder only, making search efficient using a relatively small number of anchor items, and a CUR matrix factorization. While ANNCUR's one-time selection of anchors tends to approximate the cross-encoder distances on average, doing so forfeits the capacity to accurately estimate distances to items near the query, leading to regret in the crucial end-task: recall of top-$k$ items. In this paper, we propose ADACUR, a method that adaptively, iteratively, and efficiently minimizes the approximation error for the practically important top-$k$ neighbors. It does so by iteratively performing $k$-NN search using the anchors available so far, then adding these retrieved nearest neighbors to the anchor set for the next round. Empirically, on multiple datasets, in comparison to previous traditional and state-of-the-art methods such as ANNCUR and dual-encoder-based retrieve-and-rerank, our proposed approach ADACUR consistently reduces recall error—by up to 70% on the important $k = 1$ setting—while using no more compute than its competitors.

## 1 Introduction

$k$-nearest neighbor ($k$-NN) search is a core subroutine of a variety of tasks in NLP such as entity linking (Wu et al., 2020), passage retrieval for QA (Karpukhin et al., 2020), and more generally, in retrieval-augmented machine learning models (Guu et al., 2020; Izacard et al., 2023). For many of these applications, the state-of-the-art similarity function is a cross-encoder that directly outputs a scalar similarity score after jointly encoding a

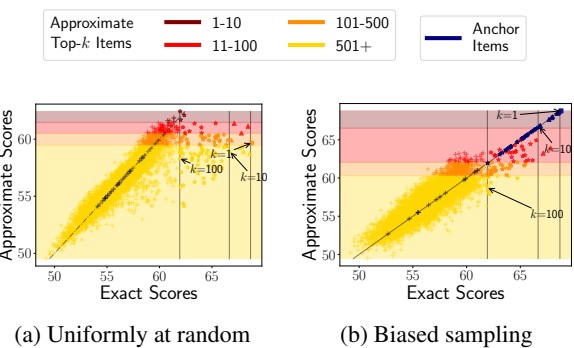

(a) Uniformly at random   (b) Biased sampling

Figure 1: Exact versus approximate cross-encoder scores (computed using ANNCUR) of all items for a test-query in domain=YuGiOh. ANNCUR incurs high approximation error on $k$-NN items wrt exact scores when using 50 anchor items sampled uniformly at random (Fig. 1a). In contrast, sampling 50 anchor items with probability proportional to exact cross-encoder scores (Fig. 1b) significantly improves approximation of top-scoring items.

given query-item pair. However, computing a *single* query-item score using a cross-encoder requires a forward pass of the model which can be computationally expensive as cross-encoders are typically parameterized using deep neural models such as transformers (Vaswani et al., 2017). For this reason, $k$-NN search with a cross-encoder typically involves retrieving candidate items using additional models such as dual-encoders or BM25 (Robertson et al., 2009), followed by re-ranking items using the cross-encoder (Logeswaran et al., 2019; Zhang and Stratos, 2021; Qu et al., 2021). However, the accuracy of such retrieve-and-rerank approaches is upperbound by the recall of first-stage retrieval and may require computationally expensive distillation-based training of dual-encoders to improve recall.

Recent work by Yadav et al. (2022) proposed ANNCUR, a CUR factorization (Mahoney and Drineas, 2009) based method, that approximates cross-encoder using dot-product of latent query and item embeddings and performs $k$-NN retrieval using approximate scores followed by optionally re-ranking retrieved items using exact cross-

encoder scores. The latent item embeddings are computed by comparing each item against a set of anchor queries, and the latent query embedding is computed using the query's cross-encoder scores against a fixed set of anchor items. As shown in Figure 1, when ANNCUR selects the anchor items uniformly at random (Fig 1a), it incurs higher approximation error on the top-scoring items than the rest of the items, resulting in poor $k$-NN recall, and including some $k$-NN items as part of anchor items (Fig. 1b) can significantly improve approximation error for top-scoring items.

In this work, we propose ADACUR, a search strategy that improves $k$-NN search recall by improving the approximation of top-scoring items. ADACUR performs retrieval over multiple rounds, retrieving the first batch of items either uniformly at random or using heuristic or auxiliary models such as dual-encoder or BM25 to get a first coarse approximation of item scores for the test query. In subsequent rounds, it alternates between a) performing retrieval using approximate scores and scoring retrieved items using cross-encoder and b) using all retrieved items as anchor items to improve the approximation and hence retrieval of relevant items in the subsequent rounds. Our proposed approach provides significant improvements in $k$-NN search recall over ANNCUR and dual-encoder based retrieve-and-rerank approaches when performing $k$-NN search with cross-encoder models trained for the task of entity linking and information retrieval.

## 2 Proposed Method: ADACUR

**Task** Given a scoring function $f_\theta \colon \mathcal{Q} \times \mathcal{I} \to \mathbb{R}$ that maps a query-item pair to a scalar score, and a query $q \in \mathcal{Q}$, the $k$-nearest neighbor search task is to retrieve top-$k$ scoring items from a fixed item set $\mathcal{I}$ according to the given scoring function $f_\theta$.

### 2.1 ADACUR: Offline Indexing of Items

The indexing step of ADACUR involves using the cross-encoder model ($f_\theta$) to score each item against a fixed set of $k_q$ anchor/train queries ($\mathcal{Q}_\text{train}$), to get score matrix $R_\text{anc}$

$$R_\text{anc}(q,i) = f_\theta(q,i), \ \forall (q,i) \in \mathcal{Q}_\text{train} \times \mathcal{I}$$

Each column of $E^\mathcal{I} := R_\text{anc} \in \mathbb{R}^{k_q \times |\mathcal{I}|}$ corresponds to a $k_q$-dimensional latent item embedding.

### 2.2 ADACUR: Test-time inference

The baseline method ANNCUR computes the latent test-query embedding $e_{q_\text{test}}$ using $C_{q_\text{test}}$, a

---

**Algorithm 1** ADACUR $k$-NN Search

1: **Input:** $(q_\text{test}, R_\text{anc}, N_\mathcal{R}, \mathcal{B}_\text{CE}, \mathcal{A})$
2: $q_\text{test}$: Test query
3: $R_\text{anc}$: Matrix containing CE scores between $\mathcal{Q}_\text{train}$ and $\mathcal{I}$
4: $\mathcal{B}_\text{CE}$: Total cross-encoder (CE) call budget.
5: $\mathcal{A}$: Algorithm to use for selecting (anchor) items.
6: $N_\mathcal{R}$: Number of iterative search rounds
7: **Output:** $\hat{S}$: Approximate scores of $q_\text{test}$ with all items, $\mathcal{I}_\text{anc}$: Retrieved (anchor) items with CE scores in $C_\text{test}$.

8: $k_s \leftarrow \mathcal{B}_\text{CE}/N_\mathcal{R}$    $\triangleright$ Num. of items to sample per round
9: $\mathcal{I}_\text{anc} \leftarrow \text{INIT}(\mathcal{I}, k_s)$    $\triangleright$ Initial set of anchor items
10: $C_\text{test} \leftarrow [f_\theta(q_\text{test}, i)]_{i \in \mathcal{I}_\text{anc}}$   $\triangleright$ CE scores of $\mathcal{I}_\text{anc}$ for $q_\text{test}$
11: **for** $j \leftarrow 2$ to $N_\mathcal{R}$ **do**
12:    $U \leftarrow R_\text{anc}[\mathcal{I}_\text{anc}]^\dagger$    $\triangleright U \in \mathbb{R}^{|\mathcal{I}_\text{anc}| \times |\mathcal{Q}_\text{train}|}$
13:    $\hat{S}^{(j)} \leftarrow C_\text{test} \times U \times R_\text{anc}$   $\triangleright$ Update approx. scores
14:    $\mathcal{I}_\text{anc}^{(j)} \leftarrow \text{SAMPLEITEMS}(\mathcal{A}, k_s, \mathcal{I}_\text{anc}, \hat{S}^{(j)})$
15:    $\mathcal{I}_\text{anc} \leftarrow \mathcal{I}_\text{anc} \cup \mathcal{I}_\text{anc}^{(j)}$
16:    $C_\text{test} \leftarrow C_\text{test} \oplus [f_\theta(q_\text{test}, i)]_{i \in \mathcal{I}_\text{anc}^{(j)}}$    $\triangleright$ Update $C_\text{test}$
17: $U \leftarrow R_\text{anc}[\mathcal{I}_\text{anc}]^\dagger$    $\triangleright U \in \mathbb{R}^{|\mathcal{I}_\text{anc}| \times |\mathcal{Q}_\text{train}|}$
18: $\hat{S} \leftarrow C_\text{test} \times U \times R_\text{anc}$    $\triangleright$ Compute approx. scores
19: **return** $\hat{S}, \mathcal{I}_\text{anc}, C_\text{test}$

---

**Algorithm 2** SAMPLEITEMS

1: **Input:** $(\mathcal{A}, k_s, \mathcal{I}_\textsf{mask}, S)$
2: $\mathcal{A}$: Algorithm for sampling items
3: $k_s$: Number of items to sample
4: $\mathcal{I}_\textsf{mask}$ : Set of items to mask
5: $S$: (Approximate) Scores for all items
6: **Output:** $\mathcal{I}_\textsf{select}$ : Set of sampled $k_s$ items

7: $\bar{S} \leftarrow \text{SOFTMAX}(S)$
8: $\bar{S}[\mathcal{I}_\textsf{mask}] \leftarrow 0$    $\triangleright$ Mask items in $\mathcal{I}_\textsf{mask}$
9: **if** $\mathcal{A} = \textsf{TopK}$ **then**
10:    $\mathcal{I}_\textsf{select} \leftarrow \text{TOPK}(\bar{S}, k_s)$
11: **else if** $\mathcal{A} = \textsf{SoftMax}$ **then**
12:    $\mathcal{I}_\textsf{select} \leftarrow k_s$ items sampled using $\bar{S}$
13: **else if** $\mathcal{A} = \textsf{Random}$ **then**
14:    $\mathcal{I}_\textsf{select} \leftarrow k_s$ items uniformly sampled from $\mathcal{I} \setminus \mathcal{I}_\textsf{mask}$
15: **return** $\mathcal{I}_\textsf{select}$

---

$|\mathcal{I}_\text{anc}|$-dimensional vector containing cross-encoder scores of $q_\text{test}$ with a set of anchor items ($\mathcal{I}_\text{anc}$) as:

$$C_{q_\text{test}} = [f_\theta(q_\text{test}, i)]_{i \in \mathcal{I}_\text{anc}}$$
$$e_{q_\text{test}} = C_{q_\text{test}} \times U$$

where $U \in \mathbb{R}^{|\mathcal{I}_\text{anc}| \times |\mathcal{Q}_\text{train}|}$ is the pseudo-inverse of $R_\text{anc}[\mathcal{I}_\text{anc}]$, the subset of columns of $R_\text{anc}$ corresponding to (anchor) items $\mathcal{I}_\text{anc}$. Finally, ANNCUR approximates the score for a query-item pair $(q_\text{test}, i)$ using dot-product of the query embedding $e_{q_\text{test}}$ and item embedding $E^\mathcal{I}[:, i]$ as

$$\hat{f}_\theta(q_\text{test}, i) = e_{q_\text{test}}^\top E^\mathcal{I}[:, i]$$

The main bottleneck at test-time inference is the number of items scored using the cross-encoder for the given test-query. For a given budget of cross-encoder calls, ANNCUR splits the budget ($\mathcal{B}_\text{CE}$)

into two parts – it uses $k_i$ cross-encoder calls to compare against anchor items (chosen uniformly at random or using heuristic methods) and retrieves $\mathcal{B}_{CE} - k_i$ items using approximate scores and re-ranks them using exact cross-encoder scores.

In contrast, our proposed approach ADACUR uses the cross-encoder call budget to adaptively retrieve and score items over $N_\mathcal{R}$ rounds, re-purposing the retrieved items as anchor items as shown in Algorithm 1. ADACUR begins by sampling the first batch of $k_s = \mathcal{B}_{CE}/N_\mathcal{R}$ (anchor) items uniformly at random. The first batch of items can also be selected using baseline retrieval methods such as BM25 and dual-encoders. In the $j^{\text{th}}$ round, the items retrieved up to round $j-1$ are used as anchor items to revise the test-query embedding, which in turn is used to update the approximate scores (line 13 in Algorithm 1). Finally, the items selected so far are masked out and the next batch of $k_s$ items in round $j$ is retrieved using the updated approximate scores in the following two ways:

- TopK: Greedily pick top-$k_s$ items according to approximate scores.

- SoftMax: Convert approximate item scores into probability using softmax and sample $k_s$ items without replacement.

Finally, ADACUR returns top-$k$ items based on exact cross-encoder scores[1] from the set of retrieved (anchor) items as approximate $k$-NN items. We refer interested readers to Appendix B.5 for a discussion on the time complexity of ADACUR.

## 3 Experiments

In our experiments, we evaluate the proposed approach and baselines on the task of finding $k$-nearest neighbor items as per a given cross-encoder. We experiment with two cross-encoders – one trained for the task of zero-shot entity linking, and another trained on information retrieval datasets.

**Experimental Setup**  We run evaluation on domains YuGiOh, StarTrek, and Military from ZESHEL–a zero-shot entity linking dataset (Logeswaran et al., 2019), and domains SciDocs and HotpotQA from BEIR–a zero-shot information retrieval benchmark (Thakur et al., 2021). We use two cross-encoder models trained on labeled training data from the corresponding benchmark and evaluate separately on each domain on the task of find-

ing $k$-NN cross-encoder items. For each ZESHEL domain, we randomly split the query set into a train/anchor set ($\mathcal{Q}_{\text{train}}$) and a test set ($\mathcal{Q}_{\text{test}}$) while for BEIR domains, we use pseudo-queries released as part of the benchmark as train/anchor queries and evaluate on queries in the official test split. We refer the reader to Table 1 for additional details.

**Baselines**  We compare our proposed approach with the following baseline retrieval methods.

**Dual-Encoders (DE)**: Query-item scores are computed using dot-product of embeddings produced by a learned deep encoder model. DE is used for initial retrieval followed by re-ranking using the cross-encoder. We report results for $DE_{BASE}$, a dual-encoder trained on training domains in the corresponding dataset, and the following two dual-encoder models trained on the target domain via distillation using the cross-encoder.

- $DE_{BERT}^{CE}$: DE initialized with BERT (Devlin et al., 2019) and trained *only* on the target domain via distillation using the cross-encoder.

- $DE_{BASE}^{CE}$: $DE_{BASE}$ model further fine-tuned on the target domain via distillation.

**ANNCUR** : $k$-NN search method proposed by Yadav et al. (2022) where anchor items are chosen uniformly at random. We additionally experiment with $ANNCUR_{DE_{BASE}}$ which uses top-scoring items retrieved using $DE_{BASE}$ as anchor items.

**Evaluation Metric**  Following the precedent set by previous work (Yadav et al., 2022), we evaluate all approaches using Top-$k$-Recall@$\mathcal{B}_{CE}$ which is defined as the fraction of $k$-NN items retrieved under test-time cost budget $\mathcal{B}_{CE}$ where the cost is defined as the number of cross-encoder calls made during inference. DE baselines will use the entire budget of $\mathcal{B}_{CE}$ calls for re-ranking retrieved items using exact cross-encoder scores, ANNCUR splits the budget between scoring anchor $k_i$ items and using exact cross-encoder scores for re-ranking $\mathcal{B}_{CE} - k_i$ retrieved items, and ADACUR use the budget to adaptively search for $k$-NN items.

For ADACUR, unless noted otherwise, we use $N_\mathcal{R} = 5$ with TopK method for retrieving items using approximate scores, and retrieve the first batch of items uniformly at random (ADACUR) or using $DE_{BASE}$ (ADACUR$_{DE_{BASE}}$). We refer readers to Appendix B for implementation details and details on training and parameterization of cross-encoder and dual-encoder models used in our experiments.

---

[1]Sorting retrieved items based on exact cross-encoder scores does not require any additional cross-encoder calls as cross-encoder scores for these items have already been computed (see line 16 in Algorithm 1).

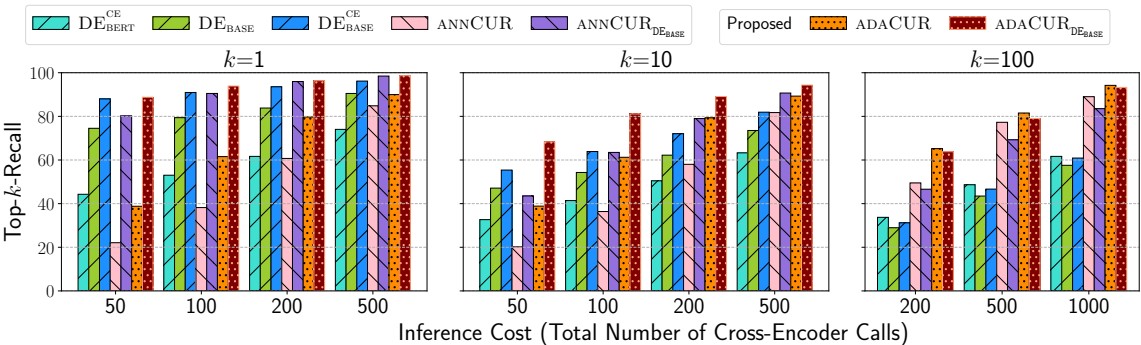

Figure 2: Top-$k$-Recall for ADACUR and baselines for domain=YuGiOh, $|\mathcal{Q}_{\text{train}}| = 500$. ADACUR consistently outperforms the corresponding ANNCUR variant and ADACUR$_{\text{DE}_{\text{BASE}}}$ outperforms all DE-based retrieve-and-rerank approaches including $\text{DE}_{\text{BASE}}^{\text{CE}}$, a DE model trained via distillation on the target domain using the cross-encoder.

## 3.1 Results

**ADACUR versus baselines** Figure 2 shows Top-$k$-Recall for ADACUR and baselines on domain=YuGiOh. ADACUR, which uses adaptively retrieved items as anchor items over $N_{\mathcal{R}} = 5$ rounds, consistently outperforms ANNCUR which selects *all* anchor items uniformly at random. ADACUR also outperforms strong DE baseline $\text{DE}_{\text{BERT}}^{\text{CE}}$ for all values of $k$ and outperforms $\text{DE}_{\text{BASE}}$ & $\text{DE}_{\text{BASE}}^{\text{CE}}$ for large values of $k = 10, 100$.

**Sampling anchor items using $\text{DE}_{\text{BASE}}$** For $k = 1, 10$, Top-$k$-Recall for both ANNCUR and ADACUR can be further improved by leveraging baseline retrieval models such as $\text{DE}_{\text{BASE}}$ for retrieving all and the first batch of anchor items respectively instead of sampling them uniformly at random. ADACUR$_{\text{DE}_{\text{BASE}}}$ always performs better than ANNCUR$_{\text{DE}_{\text{BASE}}}$ which in turn performs better than re-ranking items retrieved using $\text{DE}_{\text{BASE}}$. Thus, for a given cost budget ($\mathcal{B}_{\text{CE}}$), even when a strong baseline retrieval model such as $\text{DE}_{\text{BASE}}$ is available, using the baseline retrieval model to select the first batch of $\mathcal{B}_{\text{CE}}/N_{\mathcal{R}}$ items followed by using the proposed approach to adaptively retrieve more items over remaining $N_{\mathcal{R}} - 1$ rounds performs better than merely re-ranking $\mathcal{B}_{\text{CE}}$ top-scoring items retrieved using $\text{DE}_{\text{BASE}}$. Finally, note that ADACUR$_{\text{DE}_{\text{BASE}}}$ outperforms all baselines including $\text{DE}_{\text{BASE}}^{\text{CE}}$ which requires additional compute- and resource-intensive distillation-based fine-tuning of $\text{DE}_{\text{BASE}}$ on the target domain.

We refer the reader to Appendix C for results on other domains, training data size ($|\mathcal{Q}_{\text{train}}|$) values, oracle-based sampling experiments (§C.2), and additional results for multi-vector encoder (§C.3) and TF-IDF (§C.4) baselines.

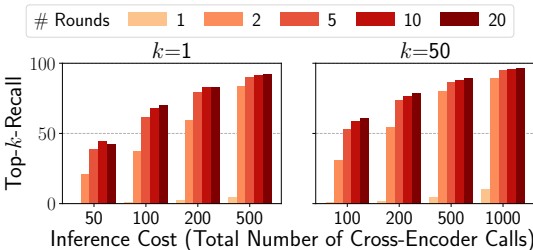

Figure 3: Top-$k$-Recall for ADACUR for different number of rounds for domain=YuGiOh, $|\mathcal{Q}_{\text{train}}| = 500$.

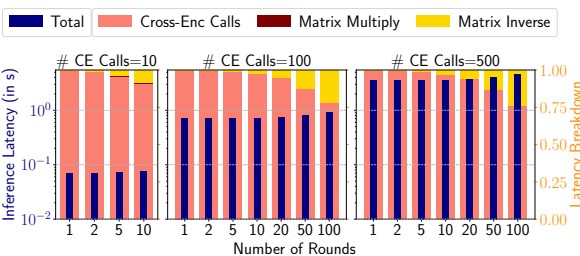

Figure 4: ADACUR inference latency versus number of rounds for domain=YuGiOh, $|\mathcal{Q}_{\text{train}}| = 500$.

**Effect of number of rounds** Figure 3 shows Top-$k$-Recall and Figure 4 shows total inference latency of ADACUR on primary y-axis for various values of the test-time cross-encoder call budget ($\mathcal{B}_{\text{CE}}$). The secondary y-axis in Figure 4 shows the fraction of total inference time spent on each one of the three main steps in Algorithm 1 – (a) computing $U$ as pseudo-inverse of $R_{\text{anc}}[\mathcal{I}_{\text{anc}}]$ (line 12), (b) updating approximate scores for all items (line 13), and (c) computing exact cross-encoder scores for retrieved items (line 16). Since ADACUR selects items uniformly at random in the first round, ADACUR with $N_{\mathcal{R}} = 1$ performs poorly as it simply returns a subset of items sampled uniformly at random. As expected, Top-$k$-Recall for ADACUR increases with the number of rounds and saturates

at around 5-10 rounds while incurring negligible overhead in inference latency. As shown in Figure 4, cross-encoder score computation is the main bottleneck in test-time inference, taking $\sim$7ms per score[2]. Increasing $N_{\mathcal{R}}$ to large values such as 100 can incur up to 25% overhead with step (a) contributing significantly to this overhead. Although the time taken by matrix multiplication in step (b) is *linear* in the number of items in the domain, we observe that it is a negligible fraction of overall latency on GPUs even for domain=HotpotQA with 5 million items (see Figure 5) as GPUs can enable significant speedup even for brute-force computation of this step.

## 4 Conclusion

In this paper, we presented an adaptive search strategy that incrementally builds a query embedding to approximate cross-encoder scores and performs $k$-NN search using approximate scores over several rounds. Our approach is designed to reduce approximation error for the top-scoring items and hence improves $k$-NN search recall when retrieving items based on the approximate scores. We perform an in-depth empirical analysis of the proposed approach in terms of both retrieval quality and efficiency.

## Limitations

Building the index for the ADACUR is more expensive than the traditional dual-encoder index due to the computation of dense cross-encoder scores matrix (see §2.1). We have successfully run our approach on up to 5 million items, but scaling to billions of items is an interesting direction for future work. Dual-encoder-based retrieve-and-rerank baseline approaches can benefit from training the dual-encoder on multiple domains. It is not clear if data from multiple domains can be leveraged to improve performance of the proposed approach on a given target domain; although in any case, cross-encoders tend to be more robust to domain shift than using only dual-encoders for retrieval.

## Ethics Statement

In this paper we tackle the task of finding $k$-nearest neighbor items for a given query when query-items scores are computed using a black-box similarity function such as a cross-encoder model. The cross-encoder scoring function may have certain biases

and error tendencies, and it is unclear if our proposed method to approximate cross-encoder scores exacerbates or mitigates such biases.

## Acknowledgments

We thank members of UMass IESL for helpful discussions and feedback. This work was supported in part by the Center for Data Science and the Center for Intelligent Information Retrieval, in part by the National Science Foundation under Grant No. NSF1763618, in part by the Chan Zuckerberg Initiative under the project "Scientific Knowledge Base Construction", in part by International Business Machines Corporation Cognitive Horizons Network agreement number W1668553, in part by Amazon Digital Services, and in part using high-performance computing equipment obtained under a grant from the Collaborative R&D Fund managed by the Massachusetts Technology Collaborative. Any opinions, findings, conclusions, and recommendations expressed in this material are those of the authors and do not necessarily reflect those of the sponsor(s).

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

# A  Appendix A

# B  Training and Implementation Details

All the code for reproducing experiments is available at https://github.com/iesl/anncur.

## B.1  Training Cross-Encoder Models

In our experiments, we use [EMB]-CE, a cross-encoder model variant proposed by Yadav et al. (2022) that jointly encodes a query-item pair and computes the final score using dot-product of contextualized query and item embeddings extracted after joint encoding.

### B.1.1  ZESHEL Dataset

ZESHEL dataset is a zero-shot entity linking containing a set of 16 domains, each containing a disjoint set of items (entities). Each domain contains a set of queries (mention) paired with their ground-truth items (entities). For ZESHEL, we use the cross-encoder model checkpoint[3] released by Yadav et al. (2022). The cross-encoder model was trained by first training a dual-encoder model on ZESHEL training data using hard negatives, and then training a cross-encoder model for the task of zero-shot entity-linking on all eight training domains using cross-entropy loss with ground-truth

---

[3]https://huggingface.co/nishantyadav/emb_crossenc_zeshel

| Dataset | Domain | $|\mathcal{I}|$ | $(|\mathcal{Q}_{\text{train}}|, |\mathcal{Q}_{\text{test}}|)$ Splits | Train Query ($\mathcal{Q}_{\text{train}}$) Type |
|---|---|---|---|---|
| ZESHEL | YuGiOh | 10,031 | (100/3274), (500/2874), (2000/1374) | Real Queries |
| ZESHEL | StarTrek | 34,430 | (100/4127), (500/3727), (2000/2227) | Real Queries |
| ZESHEL | Military | 104,520 | (100/2300), (500/1900), (2000/0400) | Real Queries |
| BEIR | SciDocs | 25,657 | (1000/1000) | Pseudo-Queries |
| BEIR | HotpotQA | 5,233,329 | (1000/1000) | Pseudo-Queries |

Table 1: Statistics on the number of items ($\mathcal{I}$) and the number of queries in train and test splits for each domain. The train-query ($\mathcal{Q}_{\text{train}}$) split refers to queries used for distilling dual-encoder models or for indexing items using ADACUR and ANNCUR. For ZESHEL domains, we create train-test splits by splitting the queries in each domain uniformly at random and test with three different splits by putting 100, 500, or 2000 queries in train split. For BEIR domains, we use pseudo-queries released as part of the benchmark as train queries ($\mathcal{Q}_{\text{train}}$) and run $k$-NN evaluation on test queries from the *official* test split (as per BEIR benchmark) of these domains. For HotpotQA, we use the first 1K queries out of a total of 7K test queries and we use all 1K test queries for SciDocs.

entity and negative entities mined using the dual-encoder. We refer readers to Yadav et al. (2022) for further details on cross-encoder training.

We perform $k$-NN experiments on domains YuGiOh, StarTrek, and Military from ZESHEL of which only Military was part of the training data used to train the cross-encoder model and YuGiOh and StarTrek are part of the original ZESHEL test domains and the cross-encoder model was *not* trained on these domains.

### B.1.2 BEIR

We follow the training setup used by Hofstätter et al. (2020). We first train three teacher cross-encoders initialized with albert-large-v2 (Lan et al., 2020), bert-large-whole-word-masking, and bert-base-uncased (Devlin et al., 2019), and compute soft labels on 40 million (query, positive item, negative item) triplets in MS-MARCO dataset (Bajaj et al., 2016). We then train our cross-encoder model parameterized using a 6-layer MINI-LM model (Wang et al., 2020) via distillation using average scores of the three teacher models as target signal and minimizing mean-square-error between predicted and target scores. We use training scripts available as part of sentence-transformer[4] repository to train the cross-encoder model and use a dot-product based scoring mechanism for cross-encoders proposed by Yadav et al. (2022)..

### B.2 Training Dual-Encoder Models

### B.2.1 ZESHEL dataset

We report results for DE baselines as reported in Yadav et al. (2022). The DE models were initialized using bert-base-uncased and contain separate query and item encoders, thus containing a total of

$2 \times 110M$ parameters. We refer readers to Yadav et al. (2022) for details related to the training of all DE model variants on ZESHEL dataset.

### B.2.2 BEIR benchmark

For BEIR domains, we use a dual-encoder model[5] released as part of sentence-transformer repository as $\text{DE}_{\text{BASE}}$. This dual-encoder model was initialized using distillbert-base (Sanh et al., 2019) and trained on MS-MARCO dataset. This $\text{DE}_{\text{BASE}}$ is *not* trained on target domains SciDocs and HotpotQA used for running $k$-NN experiments.

We finetune $\text{DE}_{\text{BASE}}$ via distillation on the target domain to get $\text{DE}_{\text{BASE}}^{\text{CE}}$ model. Given a set of training queries $\mathcal{Q}_{\text{train}}$ from the target domain, we retrieve top-100 or top-1000 items for each query, score the items with the cross-encoder model, and train the dual-encoder by minimizing cross-entropy loss between predicted query-item scores (using DE) and target query-item scores (obtained using cross-encoder). Training a $\text{DE}_{\text{BASE}}^{\text{CE}}$ with 1K queries and 100 or 1000 items per query takes around 2 hrs and 10 hrs respectively on an Nvidia RTX8000 GPU with 48GB memory. We train $\text{DE}_{\text{BASE}}^{\text{CE}}$ for 10 epochs when using top-100 items per query and for 4 epochs when using top-1000 items per query using AdamW (Loshchilov and Hutter, 2019) optimizer with learning rate 1e-5.

### B.3 ANNCUR Implementation details

For ANNCUR, we report results for the optimal split of cross-encoder call budget ($\mathcal{B}_{\text{CE}}$) between scoring $k_i$ anchor items followed by retrieving $\mathcal{B}_{\text{CE}} - k_i$ items for re-ranking. We experiment with $k_i \in \{i\mathcal{B}_{\text{CE}}/10 : 1 \leq i \leq 9\}$. If the retrieved items contain a subset of anchor items for which exact

---

[4]https://github.com/UKPLab/sentence-transformers

[5]msmarco-distilroberta-base-v2:
www.sbert.net/docs/pretrained-models/msmarco-v2.html

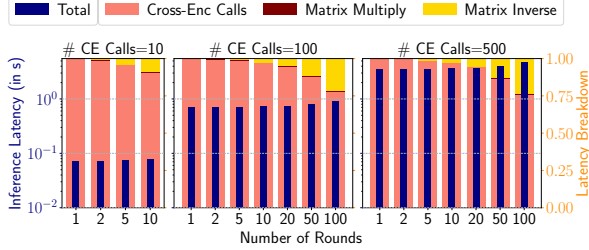
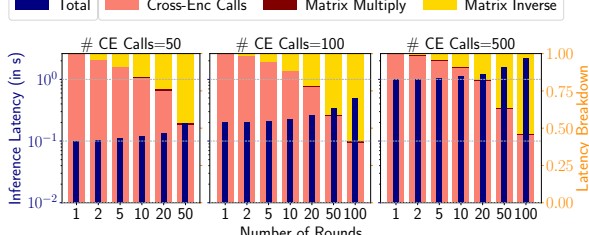

(a) Domain=Military (100K items), $|\mathcal{Q}_{\text{train}}| = 500$. Each CE call takes ∼7ms on an Nvidia 2080ti GPU for a cross-encoder parameterized using 12-layered transformer model.

(b) Domain=HotpotQA (5 Million items), $|\mathcal{Q}_{\text{train}}| = 1000$. Each CE call takes ∼2ms on an Nvidia RTX8000 GPU for a cross-encoder parameterized using 6-layered transformer model.

Figure 5: ADACUR inference latency versus number of rounds for two different domains. The primary bottleneck at inference time is the time taken to compute cross-encoder (CE) scores for query-item pairs at test time, and the overhead for ADACUR accumulates linearly as the number of rounds increases. See §B.5 for detailed discussion.

cross-encoder score has already been computed, we retrieve more than $\mathcal{B}_{\text{CE}} - k_i$ items using approximate scores and compute exact cross-encoder scores for them until we have exhausted the entire cross-encoder call budget for the re-ranking step.

## B.4 ADACUR Implementation details

For all our $k$-NN search experiments, we used Nvidia 2080ti GPUs with 12GB memory for domains YuGiOh (10K items), StarTrek (34K items), Military (100K items), and SciDocs (25K items), and we used Nvidia RTX8000 GPUs with 48GB memory for HotpotQA (5 million items).

For HotpotQA, we restrict our $k$-NN search to top-10K items wrt $\text{DE}_{\text{BASE}}$ for ADACUR$_{\text{DE}_{\text{BASE}}}$. For ZESHEL domains and SciDocs, we do not use any such heuristic and search over all the items in the corresponding domain.

## B.5 Time Complexity of ADACUR

The offline indexing step for ADACUR takes $\mathcal{O}(k_q|\mathcal{I}|\mathcal{C}_{f_\theta})$ time as it involves computing exact cross-encoder scores for all $|\mathcal{I}|$ items in the target domain against $k_q$ anchor queries, and computing each cross-encoder score takes $\mathcal{C}_{f_\theta}$ units of time.

At test time, we are given a budget $\mathcal{B}_{\text{CE}}$ on the number of cross-encoder calls. Each one of the $N_\mathcal{R}$ rounds during inference with ADACUR involves approximating all item scores for the test query ($q_{\text{test}}$) followed by sampling the next batch of $k_s = \mathcal{B}_{\text{CE}}/N_\mathcal{R}$ items using the updated approximate scores. In the $j^{th}$ round, the score approximation step involves computing the pseudo-inverse of a $k_q \times jk_s$-dimensional matrix (line 12 in Algo. 1), which takes $\mathcal{O}(\mathcal{C}_{\text{inv}}^{k_q,jk_s})$ time, followed by a matrix multiplication step to compute updated approximate scores (line 13 in Algo. 1)

which takes $\mathcal{O}(\mathcal{C}_{\text{mul}}^{k_q,jk_s,|\mathcal{I}|})$ time. The time taken to update the approximate scores in each round is $\mathcal{O}(\mathcal{C}_{\text{inv}}^{k_q,jk_s} + \mathcal{C}_{\text{mul}}^{k_q,jk_s,|\mathcal{I}|})$, and the time taken to compute cross-encoder scores for the next batch of $k_s$ items is $\mathcal{O}(k_s\mathcal{C}_{f_\theta})$. Thus, the total inference latency for retrieving items over $N_\mathcal{R}$ rounds under a given budget of $\mathcal{B}_{\text{CE}}$ cross-encoder calls is

$$\mathcal{O}\Big( \sum_{j=1}^{N_\mathcal{R}} \big( k_s\mathcal{C}_{f_\theta} + \mathcal{C}_{\text{inv}}^{k_q,jk_s} + \mathcal{C}_{\text{mul}}^{k_q,jk_s,|\mathcal{I}|} \big) \Big)$$

$$= \mathcal{O}\Big( \mathcal{B}_{\text{CE}}\mathcal{C}_{f_\theta} + \underbrace{\sum_{j=1}^{N_\mathcal{R}} \big( \mathcal{C}_{\text{inv}}^{k_q,jk_s} + \mathcal{C}_{\text{mul}}^{k_q,jk_s,|\mathcal{I}|} \big)}_{\text{Overhead of ADACUR}} \Big)$$

Figure 5 shows the breakdown of ADACUR's inference latency in terms of time spent on computing cross-encoder scores, and the overhead of computing matrix inverse in line 12 and updating approximate scores by multiplying matrices in line 13 of Algorithm 1. Empirically, we observe that the primary bottleneck at inference time is the time taken to compute cross-encoder scores for query-item pairs at test time, and the overhead for ADACUR accumulates linearly as the number of rounds increases. The overhead is mostly dominated by computing pseudo-inverse (see line 12 in Algorithm 1) and this step is independent of the target domain size. The matrix multiplication step (line 13 in Algorithm 1) has a linear dependence on the number of items in the target domain but it is a negligible fraction of the overall running time as it can be significantly sped up using GPUs.

For ZESHEL domains, we use a cross-encoder parameterized using bert-base (Devlin et al., 2019), and observe that each cross-encoder call takes amortized time of ∼7ms on an Nvidia 2080ti

GPU when the scores are computed in batches of size 50. Computing each cross-encoder score sequentially i.e. with batch-size = 1 takes ~13ms per score. We did not observe any further reduction in amortized time to compute each score when increasing the batch size beyond 50.

The amortized time per cross-encoder call is approximately 6ms and 2ms for SciDocs and HotpotQA respectively when using batch size=50 and MINI-LM-based (Wang et al., 2020) cross-encoder. The difference in time per cross-encoder score for SciDocs and HotpotQA is due to the difference in average query-item pair sequence length.

## C   Additional Results and Analysis

### C.1   TopK vs SoftMax for ADACUR

Figure 6 shows Top-$k$-Recall for ADACUR on domain=YuGiOh, $|\mathcal{Q}_{\text{train}}| = 500$, when using TopK and SoftMax strategies for sampling items based on approximate scores (see §2.2 for details). TopK sampling strategy which greedily picks top-$k$ items based on approximate scores results in superior recall as compared to sampling items using softmax over approximate scores.

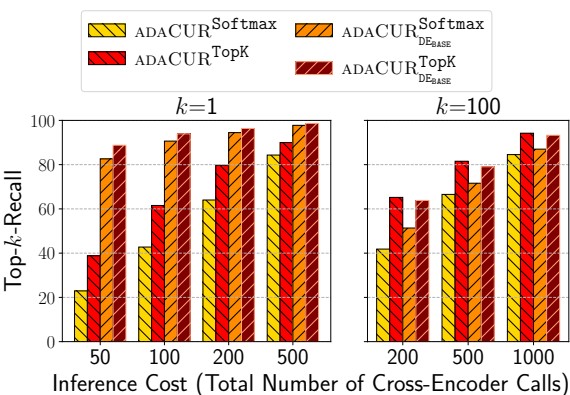

Figure 6: Top-$k$-Recall for ADACUR on YuGiOh, $|\mathcal{Q}_{\text{train}}| = 500$ for different strategies for sampling items based on approximate scores described in §2.2.

### C.2   Anchor Item Sampling with Oracle

ADACUR performs retrieval over multiple rounds using approximate cross-encoder scores and uses the items retrieved based on the approximate scores as anchor items to improve the approximation and hence retrieval in subsequent rounds. In this section, we run experiments where the anchor item sampling method has *oracle* access to exact cross-encoder scores of all items for the given test query to better understand the effect of anchor items on the approximation of cross-encoder scores and

hence subsequent retrieval based on the approximate scores. We experiment with the following strategies for sampling $k_i$ anchor items for a given test query :

- TopK$_{k_m,\epsilon}^{\mathcal{O}}$ : Mask out top-$k_m$ items wrt exact cross-encoder scores and select $k_i$ anchor items by greedily picking $(1-\epsilon)k_i$ items starting from rank $k_m + 1$, and sample remaining $\epsilon k_i$ anchor items uniformly at random.
- SoftMax$_{k_m,\epsilon}^{\mathcal{O}}$: Mask out top-$k_m$ items wrt exact cross-encoder scores and select $k_i$ anchor items by sampling $(1-\epsilon)k_i$ anchor items using softmax over exact cross-encoder scores, and sample remaining $\epsilon k_i$ anchor items uniformly at random.

For a given test-time cross-encoder call budget $\mathcal{B}_{\text{CE}}$, we select $k_i$ anchor items, compute approximate cross-encoder scores using the chosen anchor items, and then retrieve $\mathcal{B}_{\text{CE}} - k_i$ items based on the approximate scores. We experiment with $k_i \in \{i\mathcal{B}_{\text{CE}}/10 : 1 \leq i \leq 9\}$ and report results for the best budget split.

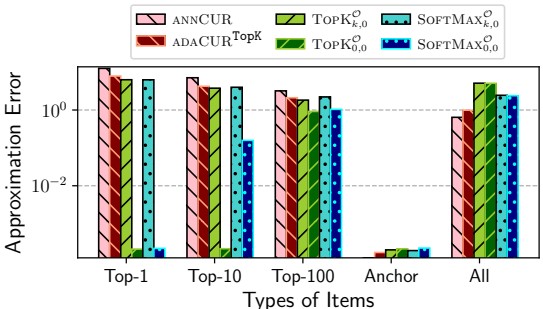

Figure 7: Average approximation error for CUR matrix factorization on test-queries for domain=YuGiOh and $|\mathcal{Q}_{\text{train}}| = 500$ when choosing $k_i = 50$ anchor items uniformly at random (ANNCUR), using oracle strategies from §C.2 and for ADACUR when sampling anchor items over five rounds. Approximation error is computed as the average of absolute difference between approximate and exact item scores.

**Effect of adding $k$-NN items to anchor items**
Figure 8a shows Top-$k$-Recall of anchor item sampling strategies TopK$_{k_m,0}^{\mathcal{O}}$ and SoftMax$_{k_m,0}^{\mathcal{O}}$ for $k_m \in \{0,k\}$, domain=YuGiOh. Sampling strategies SoftMax$_{k,0}^{\mathcal{O}}$ and TopK$_{k,0}^{\mathcal{O}}$, which mask out top-$k$ items, perform significantly worse than SoftMax$_{0,0}^{\mathcal{O}}$ and TopK$_{0,0}^{\mathcal{O}}$ respectively when searching for $k = 1, 10$ nearest neighbors. This indicates that the significant improvement in Top-1-Recall and Top-10-Recall for TopK$_{0,0}^{\mathcal{O}}$ and SoftMax$_{0,0}^{\mathcal{O}}$ sampling strategies can be attributed to the presence of top-$k$ items in the anchor item set. This is because CUR matrix

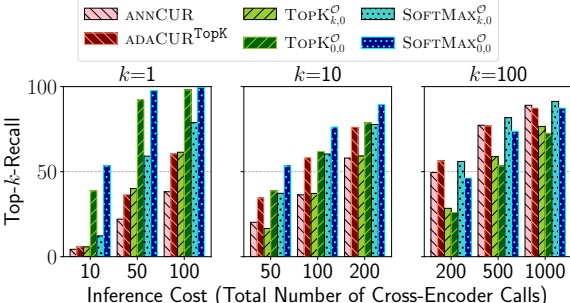

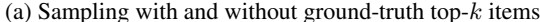

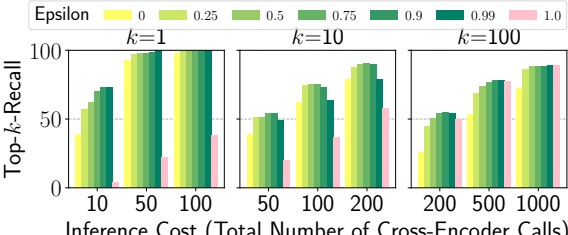

(a) Sampling with and without ground-truth top-$k$ items

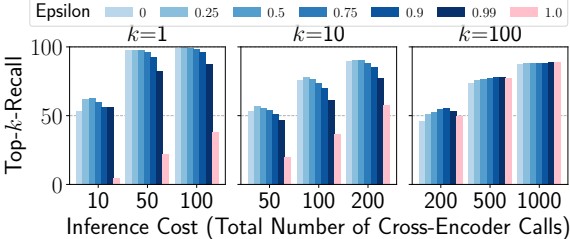

(b) Sampling using TopK strategy with exact CE scores while varying $\epsilon$, the fraction of items sampled uniformly at random

(c) Sampling using SoftMax of exact CE scores while varying $\epsilon$, the fraction of items sampled uniformly at random

Figure 8: Top-$k$-Recall for ADACUR, ANNCUR, and oracle sampling strategies (§C.2) that have oracle access to exact cross-encoder scores for all items for domain=YuGiOh, $|\mathcal{Q}_{\text{train}}| = 500$.

factorization which is used to compute the approximate scores incurs negligible approximation error on anchor items, and hence on top-$k$ items when these items are part of the anchor set as shown in figures 7 and 9. For $\text{TopK}^{\mathcal{O}}_{k,0}$ and $\text{SoftMax}^{\mathcal{O}}_{k,0}$ sampling strategies, since the top-$k$ items are *not* part of the anchor set, CUR incurs a much higher approximation error for the top-$k$ items (see examples in Figures 9a and 9b ), thus resulting in poor Top-$k$-Recall as shown in Figure 8a.

**Effect of *diversity* in anchor items** Figure 8a shows that sampling items based on softmax of exact cross-encoder scores ($\text{SoftMax}^{\mathcal{O}}_{k_m,0}$) performs better than greedily picking top-scoring items ($\text{TopK}^{\mathcal{O}}_{k_m,0}$), for both $k_m = 0, k$. The reason behind $\text{SoftMax}^{\mathcal{O}}_{k_m,0}$ performing better than $\text{TopK}^{\mathcal{O}}_{k_m,0}$ is that sampling based on softmax of exact scores yields an anchor set with a more *diverse*

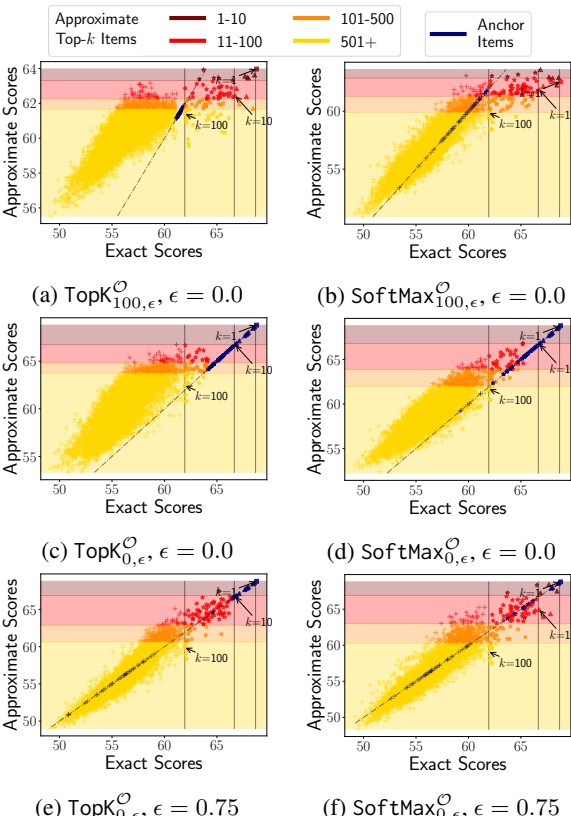

(a) $\text{TopK}^{\mathcal{O}}_{100,\epsilon}, \epsilon = 0.0$  (b) $\text{SoftMax}^{\mathcal{O}}_{100,\epsilon}, \epsilon = 0.0$

(c) $\text{TopK}^{\mathcal{O}}_{0,\epsilon}, \epsilon = 0.0$  (d) $\text{SoftMax}^{\mathcal{O}}_{0,\epsilon}, \epsilon = 0.0$

(e) $\text{TopK}^{\mathcal{O}}_{0,\epsilon}, \epsilon = 0.75$  (f) $\text{SoftMax}^{\mathcal{O}}_{0,\epsilon}, \epsilon = 0.75$

Figure 9: Scatter plot showing approximate versus exact cross-encoder scores for a query from domain=YuGiOh, when choosing $k_i = 50$ anchor items using oracle strategies from §C.2 and $|\mathcal{Q}_{\text{train}}| = 500$. Top-$k$ for $k=1,10,100$ wrt exact cross-encoder scores are annotated with text along with vertical lines, different color bands indicate the ordering of items wrt approximate scores, and anchor items are shown in blue.

score distribution whereas greedily selecting top-scoring items using exact scores results in an anchor set with items having similar cross-encoder scores. However, as shown in Figures 9c and 9d, both of these sampling strategies can result in overestimating scores for all items, even the irrelevant ones (i.e. items beyond top-$k$ items) due to insufficient representation of the irrelevant items in the anchor set. Thus retrieving based on approximated scores may struggle to retrieve relevant $k$-NN items, especially for larger values of $k$ such as $k = 100$ when the anchor items are chosen using oracle strategies such as $\text{TopK}^{\mathcal{O}}_{k_m,0}$.

Figures 9e and 9f, where $\epsilon = 75\%$ of 50 items are sampled uniformly at random, show that overestimating scores of irrelevant items can be avoided by sampling a fraction of anchor items uniformly at random to increase the diversity of the anchor item set. As shown in Figures 8b and 8c, Top-$k$-Recall for both $\text{SoftMax}^{\mathcal{O}}_{0,\epsilon}$ and $\text{TopK}^{\mathcal{O}}_{0,\epsilon}$ generally improves with an increase in $\epsilon$, the fraction of ran-

dom items in the anchor set, due to increased diversity in the anchor item set. Since $\mathrm{SoftMax}_{0,\epsilon}^{\mathcal{O}}$ already samples a diverse set of anchor items, increasing $\epsilon$ yields only marginal improvement while for $\mathrm{TopK}_{0,\epsilon}^{\mathcal{O}}$, increasing $\epsilon$ yields significant improvements due to increased diversity of the anchor set. A small drop in performance is observed for larger values of $\epsilon$ as increasing $\epsilon$ beyond a threshold results in some of the top-$k$ items to be excluded from the anchor item set. This results in a poorer approximation of scores for the missing top-$k$ items and hence poor retrieval recall as the retrieval is done using the approximate scores.

Finally, the optimal strategy for choosing the set of anchor items is the one that strikes a balance between selecting anchor items with diverse cross-encoder scores and greedily picking top-$k$ items. Our proposed strategy ADACUR improves over ANNCUR as greedily picking top-scoring items according to approximate scores to expand set of anchor items increases the likelihood of picking ground-truth $k$-NN items to be part of the anchor set, with this likelihood improving after each round with improvement in the score approximation, and ADACUR achieves diversity in the anchor items as a result of sampling items uniformly at random in the first round and due to error in the approximate scores, as shown in Figure 11.

## C.3 Comparison with Multi-Vector Models

Multi-vector models (Khattab and Zaharia, 2020; Ma et al., 2021) produce multiple embeddings for each query and item. For a given query $q$ and item $i$, the query-item score is computed using simple functions such as average similarity or sum-of-maximum similarities between the set of embeddings for query $q$ and item $i$.

Figure 10 shows Top-$k$-Recall for $\mathrm{DE_{BASE}}$, $\mathrm{DE_{BASE}^{CE}}$, $\mathrm{ADACUR_{DE_{BASE}}}$, and MUVER (Ma et al., 2021), a recent multi-vector model trained on ZESHEL dataset. For MUVER, we use the pretrained checkpoint released by Ma et al. (2021) with the *view-merging* inference strategy as described in Ma et al. (2021). While MUVER can be more accurate than $\mathrm{DE_{BASE}}$, $\mathrm{DE_{BASE}^{CE}}$ obtained by finetuning $\mathrm{DE_{BASE}}$ model on the target domain outperforms MUVER and our proposed method $\mathrm{ADACUR_{DE_{BASE}}}$ yields the best Top-$k$-Recall versus inference cost trade-offs for all values of $k$.

We would also like to note that while multi-vector models such as MUVER can be more accurate than single-embedding models such as $\mathrm{DE_{BASE}}$,

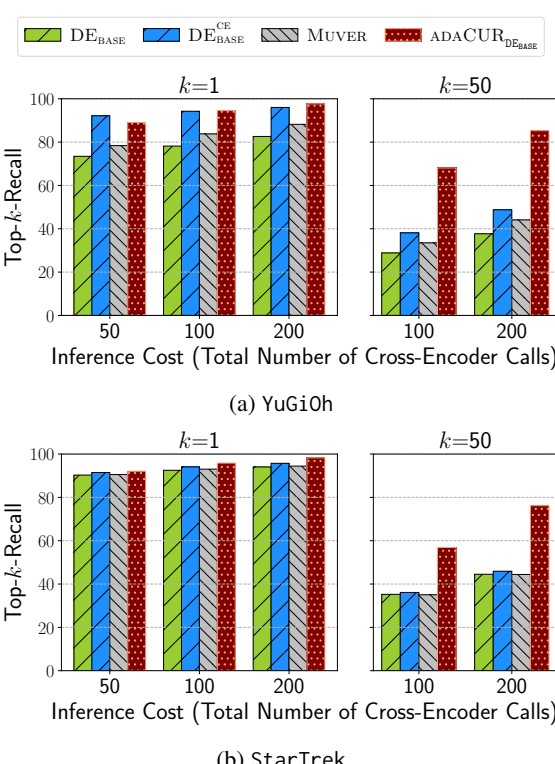

Figure 10: Top-$k$-Recall for $\mathrm{ADACUR_{DE_{BASE}}}$ and baselines including multi-vector models on ZESHEL domains, $|\mathcal{Q}_{\mathrm{train}}| = 2000$. See §C.3 for discussion.

such multi-vector models incur significant memory overhead for storing query/item embeddings. For instance, using 15 embeddings per item with 768-dimensional embeddings would take around 250GB space for 5 million items for HotpotQA.

## C.4 Results for TF-IDF baseline

TF-IDF: All queries and items are embedded using a TF-IDF vectorizer trained on item descriptions and top-$k$ items are retrieved using the dot-product of sparse query and item embeddings.

For domains in ZESHEL, we report results for TF-IDF baseline, for ANNCUR when anchor items are chosen using TF-IDF baseline (ANNCUR_{TF-IDF}), and for ADACUR when the first batch of anchor items is chosen using TF-IDF baseline (ADACUR_{TF-IDF}). Figures 13, 14, and 15 show Top-$k$-Recall for domains YuGiOh, StarTrek, and Military respectively for $|\mathcal{Q}_{\mathrm{train}}| \in \{100, 500, 2000\}$. For each baseline retrieval method, ADACUR always performs better than ANNCUR which in turn generally performs better than merely re-ranking items retrieved using the corresponding baseline retrieval method. In most cases, Top-$k$-Recall for $\mathrm{ADACUR_{DE_{BASE}}} > \mathrm{ANNCUR_{DE_{BASE}}} > \mathrm{DE_{BASE}}$, and $\mathrm{ADACUR_{TF-IDF}} > \mathrm{ANNCUR_{TF-IDF}} > \mathrm{TF-IDF}$.

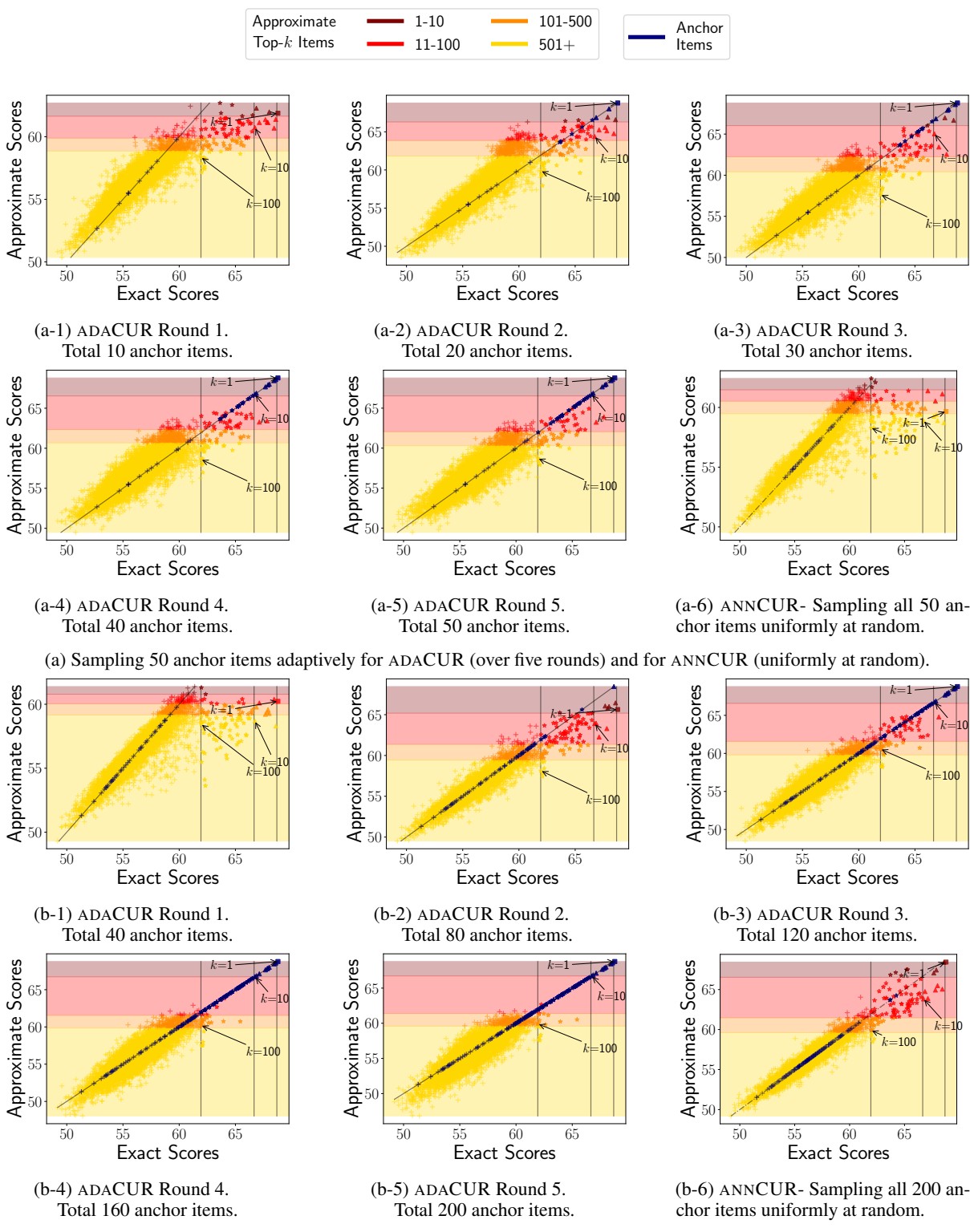

(a-1) ADACUR Round 1.
Total 10 anchor items.

(a-2) ADACUR Round 2.
Total 20 anchor items.

(a-3) ADACUR Round 3.
Total 30 anchor items.

(a-4) ADACUR Round 4.
Total 40 anchor items.

(a-5) ADACUR Round 5.
Total 50 anchor items.

(a-6) ANNCUR- Sampling all 50 anchor items uniformly at random.

(a) Sampling 50 anchor items adaptively for ADACUR (over five rounds) and for ANNCUR (uniformly at random).

(b-1) ADACUR Round 1.
Total 40 anchor items.

(b-2) ADACUR Round 2.
Total 80 anchor items.

(b-3) ADACUR Round 3.
Total 120 anchor items.

(b-4) ADACUR Round 4.
Total 160 anchor items.

(b-5) ADACUR Round 5.
Total 200 anchor items.

(b-6) ANNCUR- Sampling all 200 anchor items uniformly at random.

(b) Sampling 200 anchor items adaptively for ADACUR (over five rounds) and for ANNCUR (uniformly at random).

Figure 11: Scatter plot showing approximate versus exact cross-encoder scores for a query from domain=YuGiOh, $|\mathcal{Q}_{\text{train}}| = 500$ when choosing $k_i = 50$ and 200 anchor items with ADACUR over five rounds, and uniformly at random with ANNCUR. Top-$k$ for $k$=1,10,100 wrt exact cross-encoder scores are annotated with text, different color bands indicate the ordering of items wrt approximate scores, and anchor items are shown in blue. With ADACUR, the first batch containing anchor items in Figure 11a-1 and 11b-1 is chosen uniformly at random and in subsequent rounds, items with highest approximate scores are chosen. Note that the approximation error for top-scoring items improves significantly when the 50 anchor items are chosen adaptively (see Figure 11a-5) with the improvement being much more significant than merely increasing the number of anchor items sampled uniformly at random from 50 in Figure 11a-6 to 200 in Figure 11b-6.

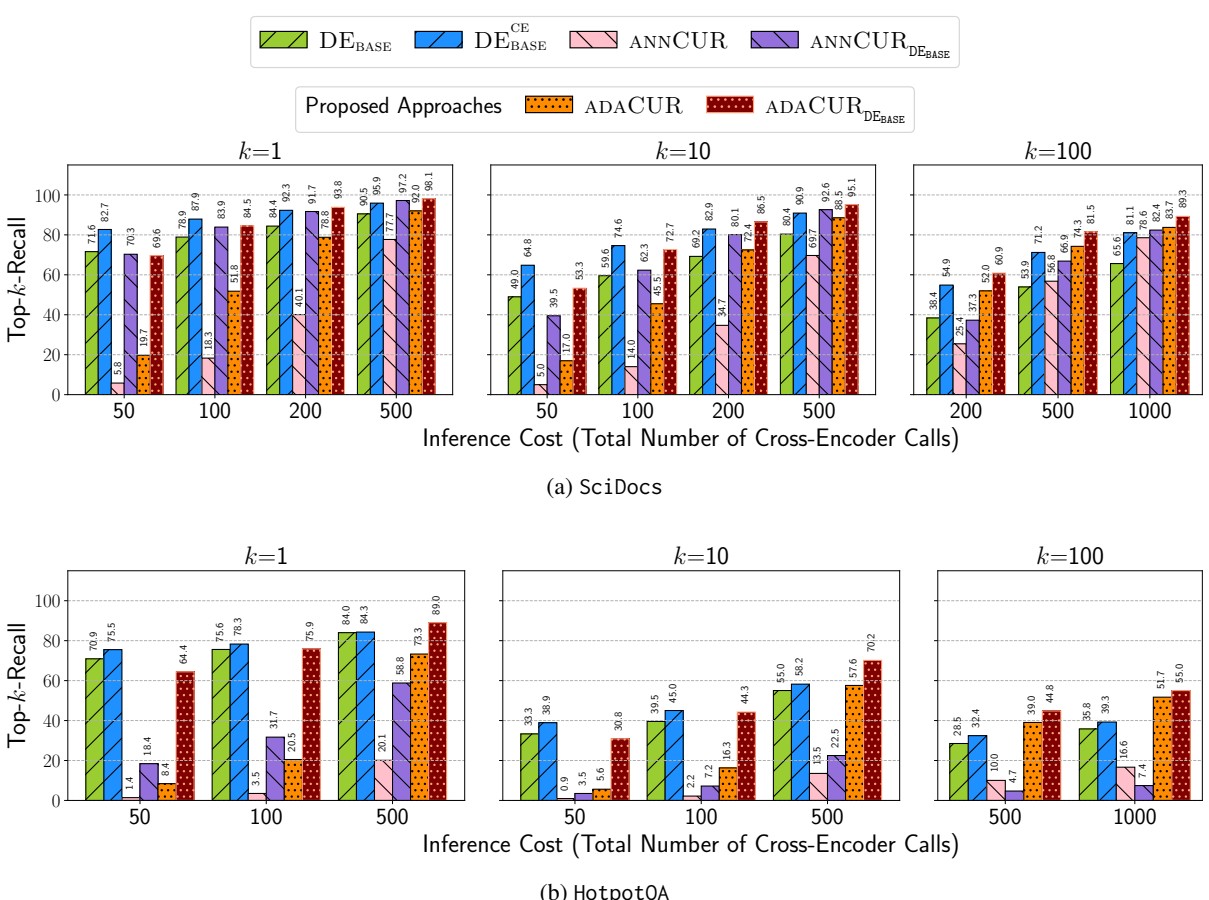

(a) SciDocs

(b) HotpotQA

Figure 12: Top-$k$-Recall for ADACUR (using ten rounds) and baselines for SciDocs and HotpotQA, $|\mathcal{Q}_{\text{train}}| = 1000$.

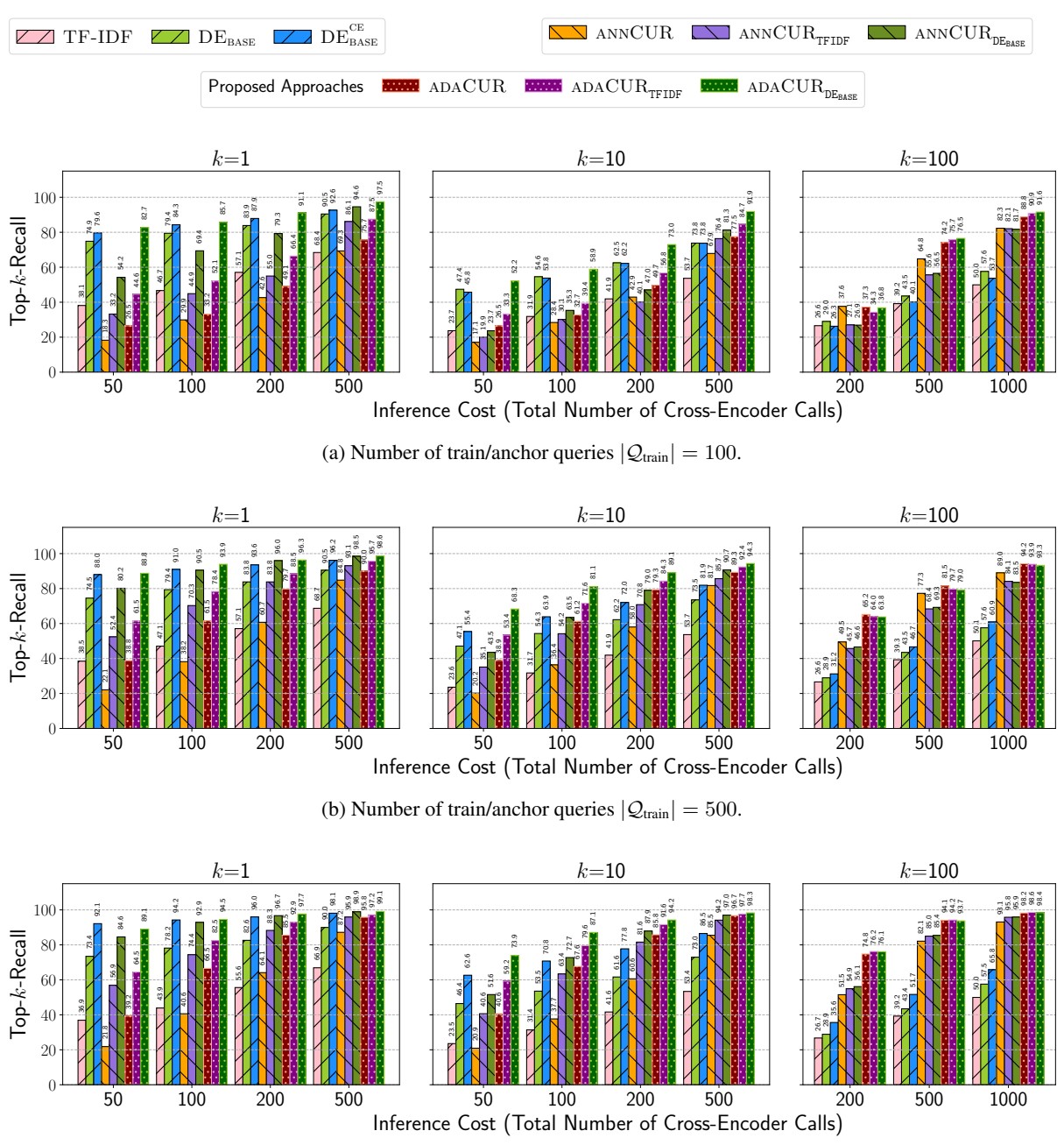

Figure 13: Top-$k$-Recall for ADACUR (using five rounds) and baselines for domain=YuGiOh. Each subfigure corresponds to a different value of the number of train/anchor queries ($|\mathcal{Q}_{\text{train}}|$).

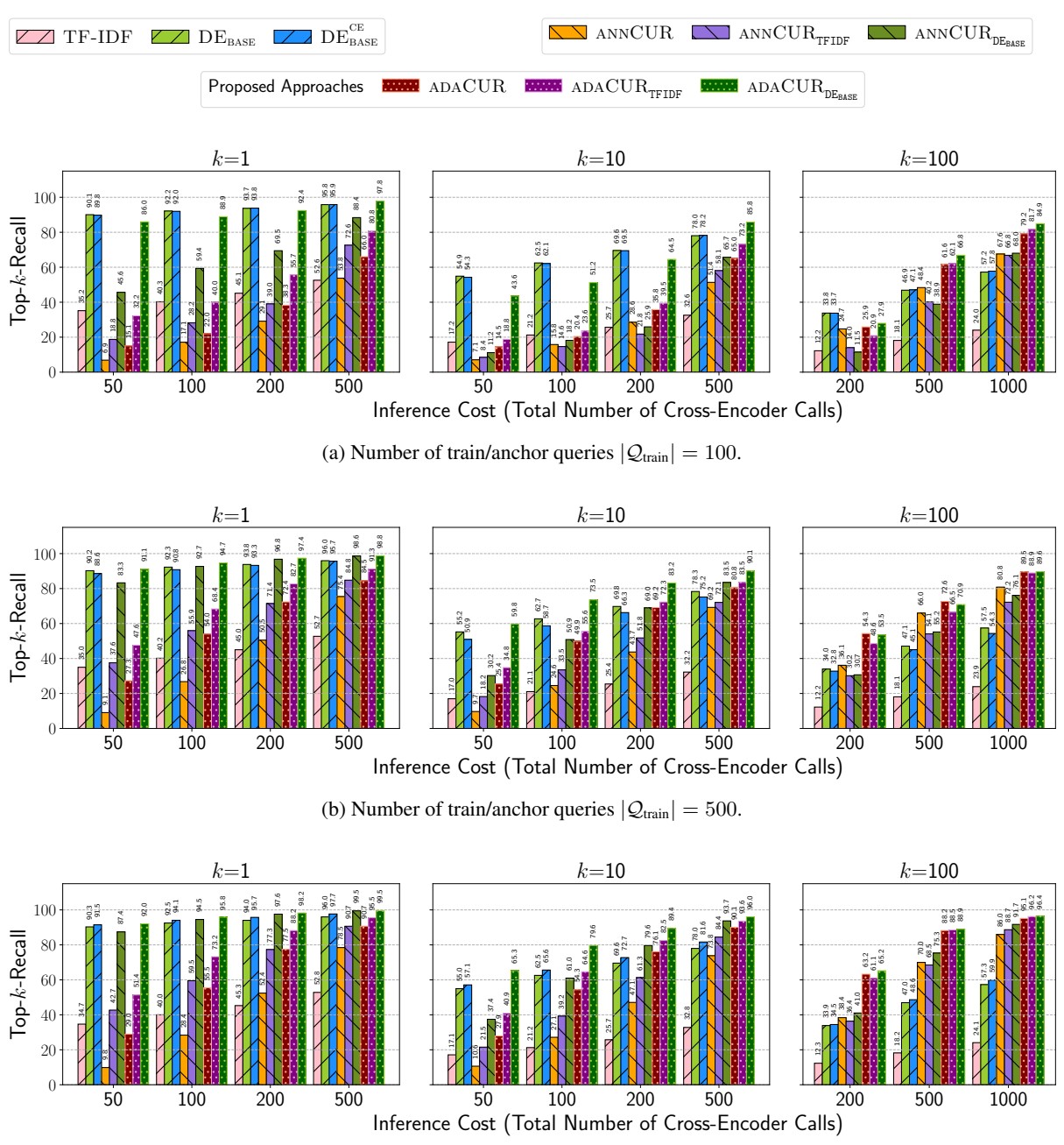

Figure 14: Top-$k$-Recall for ADACUR (using five rounds) and baselines for domain=StarTrek. Each subfigure corresponds to a different value of the number of train/anchor queries ($|\mathcal{Q}_{\text{train}}|$).

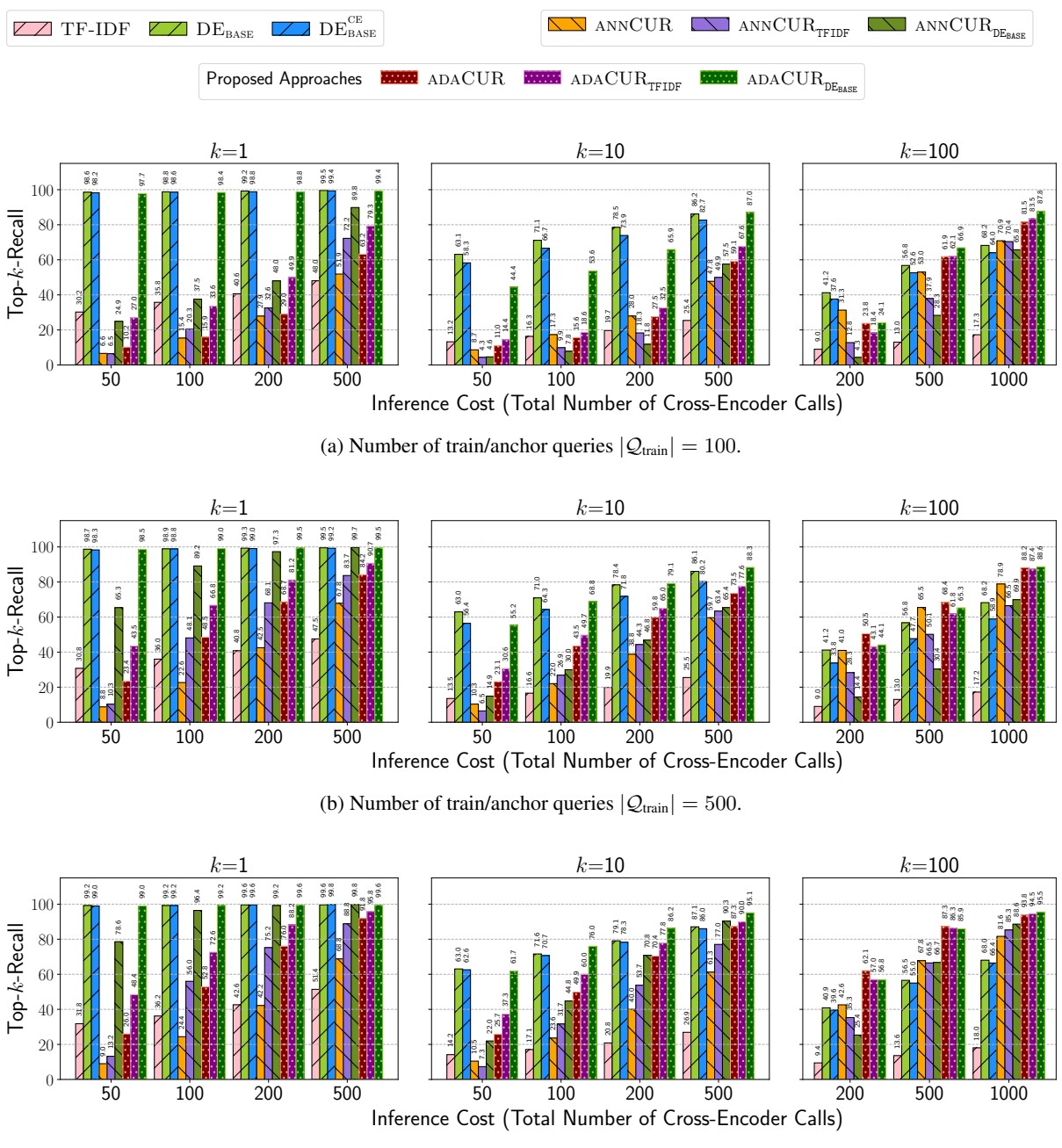

(a) Number of train/anchor queries $|\mathcal{Q}_{\text{train}}| = 100$.

(b) Number of train/anchor queries $|\mathcal{Q}_{\text{train}}| = 500$.

(c) Number of train/anchor queries $|\mathcal{Q}_{\text{train}}| = 2000$.

Figure 15: Top-$k$-Recall for ADACUR (using five rounds) and baselines for domain=Military. Each subfigure corresponds to a different value of the number of train/anchor queries ($|\mathcal{Q}_{\text{train}}|$). Note that DE$_{\text{BASE}}$ has high Top-1-Recall values as domain=Military is included in the set of train domains in ZESHEL which are used to train both DE$_{\text{BASE}}$ and the cross-encoder model.