# OpenReview forum: "Efficient k-NN Search with Cross-Encoders using Adaptive Multi-Round CUR Decomposition"
_EMNLP/2023/Conference — EMNLP 2023 Findings_

### Official Review · Reviewer_sVVR · 2023-08-05

**Soundness:** 3

**Excitement:**

4: Strong: This paper deepens the understanding of some phenomenon or lowers the barriers to an existing research direction.

**Missing References:**

1. MuVER: Improving First-Stage Entity Retrieval with Multi-View Entity Representations, EMNLP 2021.
2. Towards Better Entity Linking with Multi-View Enhanced Distillation, ACL 2023.
3. A Transformational Biencoder with In-Domain Negative Sampling for Zero-Shot Entity Linking, ACL 2022.
4. EntQA: Entity Linking as Question Answering, ICLR 2022.
5. Entity Linking via Explicit Mention-Mention Coreference Modeling, NAACL 2022.
6. TempEL: Linking Dynamically Evolving and Newly Emerging Entities, Neurips 2022.
7. Modeling Fine-grained Information via Knowledge-aware Hierarchical Graph for Zero-shot Entity Retrieval, WSDM 2023.

**Paper Topic And Main Contributions:**

AdaCUR is an extension of AnnCUR, a method of efficiently leveraging cross-encoders for KNN search in the context of entity linking, passage retrieval, etc. AdaCUR has meaningful yet incremental improvement over AnnCUR. The key insight is that while the anchor queries are selected in the offline indexing stage and considered as fixed, the anchor items may change at test-time based on the test query.

**Questions For The Authors:**

**A.** Please add experiment results of more recently proposed methods for comparison. The list of missing references below includes some candidate methods.

**B.** Not sure if I missed it, but why is the improvement when k=10 often more prominent than the improvement when k=1 and k=100? Also, why is AnnCUR even worse than TF-IDF in Figures 14 and 15 in the appendix?

**Reasons To Accept:**

Overall the CUR factorization presents an interesting view of how cross-encoder approaches can be accelerated in practice. The proposed method has proper motivation to better approximate the cross-encoder scores of the nearest items. Experiment shows AdaCUR has higher recall than AnnCUR with the same cross-encoder model.

**Reasons To Reject:**

In the experiments, the authors compared AdaCUR with AnnCUR, dual-encoder methods (probably by Zhang and Stratos, 2021; the authors didn't specify), and TF-IDF baselines. But more recent methods are not included in the experiments. Please see the missing references for a list of potential papers for comparison.

While this paper pays good attention on keeping the cost fixed, i.e. having the same number of cross-encoder inferences, real-world time costs will depend on the size of the item collection, the number of anchor queries, and the number of anchor items. When is AdaCUR favored against the retrieve-and-rerank strategy and other methods that avoid cross-encoder models, in terms of the problem size? And when is AdaCUR not favored? These are the questions that might be more interested to readers, rather than merely an improvement of the recall metric. Maybe the current writing is limited by the page length of a short paper, but I believe it could even be converted to a regular paper if the above questions are addressed.

The current paper writing is not self-contained. I have to go through the previous AnnCUR paper in order to understand what is happening.

**Reproducibility:**

4: Could mostly reproduce the results, but there may be some variation because of sample variance or minor variations in their interpretation of the protocol or method.

**Reviewer Confidence:**

4: Quite sure. I tried to check the important points carefully. It's unlikely, though conceivable, that I missed something that should affect my ratings.

**Typos Grammar Style And Presentation Improvements:**

The abstract is unnecessarily long. Without the context of anchor queries, anchor items, and CUR, the description of the shortcoming of AnnCUR and the mechanism of AdaCUR doesn't make too much sense for a reader.

The introduction section needs to be more self-contained. As of now, it never introduced the concept of anchors and CUR properly. Figure 1 would be of great insight if the CUR framework was introduced first; otherwise, the discussion of approximating the average scores well and the need for biased sampling don't have the right context.

Line 84: q_test is used here while the definition of test-time queries doesn't appear until Line 112.

Lines 103-123 are actually AnnCUR. AdaCUR starts from Line 124. I suggest changing the titles of sections 2.1 and 2.2 accordingly.

Line 126: "for a given budget B_CE of CE calls". I would really appreciate a rephrase for easier understanding: "for a given budget B_CE, the number of total CE calls".

Algorithm 1, Line 16: This plus operator is not defined in the paper; I can only guess it.

Overall, I think the writing should be reorganized.

---

> ### Author Rebuttal · Authors · 2023-08-28
>
> Thank you very much for your thoughtful suggestions, comments and time reviewing this paper. We will revise the paper to fix presentation and writing related issues pointed out by the reviewer.
>
> ### **Re: Comparison with suggested baselines**
>
> Thanks for the suggested references.
>
> In our work, we make following assumptions about models/data:
> - We are given a trained cross-encoder model
> - We are given data from target domain to index
> - Optionally, we are given a dual-encoder model trained on _same_ task and source domains as cross-encoder
> - Do not assume access to source domain data (which was used to train cross-encoder)
> - Do not assume any special structures or relations amongst queries or amongst items
>
> We report results for additional experiments with models from _MuVER_ (Ma et al. 2021) below. To the best of our understanding, references other than Ma et al. (2021)
> - either focus solely on the task of entity linking and do not apply to the general task of k-NN search with a given cross-encoder model (which is the focus of our work)
> - or require training both dual-encoders or cross-encoders _jointly_, thus are not compatible with our setting where we assume that we are given a trained cross-encoder.
>
> We present a more detailed response for each paper below.
>
> * _MuVER: Improving First-Stage Entity Retrieval with Multi-View Entity Representations, EMNLP 2021_
>
>   * We run kNN search evaluation using the MuVER checkpoint released by Ma et al. (2021).
>   * For brevity, we show Top-1-Recall for MuVER and other models (from Fig 2 in our paper) at different test-time costs for doman=YuGiOh.
>    * While MuVER performs better than base dual-encoder (DE), it performs worse as compared to dual-encoders finetuned on the target domain and AdaCUR, our proposed approach. We will add these additional results to the paper.
>
>
> | Model\Inference Cost | 50     | 100   | 200   |
> | ---           | ---           | ---   | ---   |
> | Base DE       		|  74.5	| 79.4	| 83.8  |
> | MuVER         		|  78.3	| 83.5  	| 88.1  |
> | Distilled DE 	 	|  88.0	| 90.9	| 93.6  |
> | AdaCUR (Ours) 	|  88.8	| 93.9	| 96.4  |
>
>
>
>  * *Memory overhead for MuVER and multi-vector models*
>
>     - While models that generate multiple embeddings per query and item (eg. MuVER) can be more accurate than models that generate a single embedding (eg. dual-encoder), such multi-vector models incur significant memory overhead for storing the query and item representations, and require special index structure for efficient inference even at moderate scale.
>     - For instance, using 15 embeddings per item, as used by Ma et al. (2021), would take around 250GB space to store 768-dim embeddings for 5 million items for HotpotQA domain.
>     - Moreover, training such multi-vector models also require more compute resources and time as compared to training single-embedding dual-encoder baselines.
>     - This makes training and using such multi-vector models in order to perform first stage retrieval for a given cross-encoder model a less attractive option.
>
>
> **As mentioned before, to the best of our understanding, following references are _not suitable_ baselines for our work. We present a detailed response for each paper below.**
>
>   * _Modeling Fine-grained Information via Knowledge-aware Hierarchical Graph for Zero-shot Entity Retrieval_
>     * This paper focuses on entity linking and the proposed ideas do not apply to general kNN search/information retrieval settings.
>     * In particular, this paper extracts subject-verb-predicate triplets from surrounding text to improve mention embedding and such an approach can not be used for improving query embedding general k-NN search setting such as passage retrieval setting.
>
>   * _EntQA: Entity Linking as Question Answering_
>     * This paper links mentions in a document to entities by first shortlisting some entities and then finding potential mention span for each shortlisted entity.
>     * Such an approach cannot be generalized to a more general kNN search setting. For eg, it cannot be applied to finding documents/passages for a given query.
>
>   * _Entity Linking via Explicit Mention-Mention Coreference Modeling (NAACL 2022)_
>     * This paper assumes access to mention-mention coreference decisions while training the dual-encoder model.
>     * Such method can not be used for training dual-encoders in the general information retrieval settings eg. passage retrieval given a query.
>
>   * _TempEL: Linking Dynamically Evolving and Newly Emerging Entities (NeurIPS 2022)_
>     * The primary contribution of this paper is a new dataset to study how entities evolve and emerge in a KB.
>     * To the best of our understanding, it _does not propose_ a new approach/model for entity linking.
>
>   * _Towards Better Entity Linking with Multi-View Enhanced Distillation (Liu et al. ACL 2023)_
>     * Thanks for pointing us to this paper. We missed this as it was published only a month before the EMNLP deadline.
>     * Nevertheless, models proposed in Liu et al. (2023) paper _may not be suitable baseline_ as best results in Liu et al. (2023) correspond to a setting where
>       * both cross-encoder and dual-encoder are trained jointly – Not compatible with our setting where we are given a trained cross-encoder
>       * cross-encoder trained on multi-view data – thus may _not_ work well with general cross-encoders.
>     * In fact, as shown in ablations in Liu et al. (2023), using a cross-encoder trained separately and without multi-view data results in a dual-encoder with entity linking recall@64 of 88.5%.
>       * In our work, we use the dual-encoder checkpoint released by Yadav et al. (2022) which has entity linking recall@64 ~88%, and we also compare with stronger baselines such dual-encoders finetuned on the target domain.
>       * Also note, that our evaluation metrics use _top-k items/entities wrt cross-encoder_ as ground-truth for evaluation while evaluation in Liu et al. (2023) focus on entity linking where the goal is finding the single ground-truth entity (item) which is specified as part of the dataset annotation.
>
>
>   * _A Transformational Biencoder with In-Domain Negative Sampling for Zero-Shot Entity Linking (Sun et al. ACL 2022)_
>     * The ideas proposed in this paper are geared towards improving performance of a dual-encoder in _zero-shot domain transfer_ setting.
>     * The transformational biencoder idea may not yield any improvement for our use-case when we finetune the dual-encoder on the target domain and then use it for kNN search for test-query on the _same target_ domain.
>     * Ideas from Sun et al. (2022) could be used to train better base dual-encoder models on the source data which can be then finetuned on the target data.
>     * However, for fair comparison with prior work, in this work we use and finetune publicly available dual-encoder models released by Yadav et al. (2022) for ZeShEL data and dual-encoders available on huggingface for BeIR dataset which do not use the transformational biencoder idea.
>
>
> ### **When is AdaCUR favored over retrieve-and-rerank?**
>
> Let’s begin by reviewing consistent empirical trends in our results:
>
> * For k=100, AdaCUR consistently provides improvement over retrieve-and-rerank and all other baselines including strong dual-encoder based fine-tuning approaches.
> * For k=1, AdaCUR consistently outperforms AnnCUR under all settings and outperforms strong dual-encoder based retrieve-and-rerank approaches at inference cost budgets >= 100.
>
> * We agree that the total number of items and number of anchor queries affect the performance of AdaCUR as well as baselines.
>     * For this reason, we presented results for multiple datasets/domains containing varying number of items (from 10K items in domain=YuGiOh to 5 Million items in HotpotQA dataset).
>     * We also show results for different number of anchor queries in Figure 13, 14 and 15 for domains in ZeShEL dataset, and as expected performance of AdaCUR over retrieve-and-rerank baselines generally improves with the number of anchor/training queries.
>
> * A few combinations where retrieve-and-rerank approaches perform better than AdaCUR
>     * In Fig 14a, when AdaCUR uses _only_ 100 anchor queries for indexing the items, dual-encoder based retrieve-and-rerank performs better than AdaCUR at small cost budgets (e.g. cost=50).
>     * But even in these settings, AdaCUR _closely matches or outperforms_ dual-encoders at larger cost budgets.
>
>
> **Why is the improvement when k=10 often more prominent than the improvement when $k=1$ and $k=100$?**
>
>   * We would like to note that AdaCUR offers consistent improvements over baselines for all values of $k=1, 10, 100$ as shown in Figure 2.
>   * The improvements for $k=10$ might appear to be more prominent than for $k=1$ and $k=100$ as the trends for relative improvements over baselines vary by baseline and are also affected by the inference cost.
>     * For instance, the improvements provided by $\text{AdaCUR}_{\text{DE-Base}}$ over dual-encoder (DE) baselines consistently increase as we go from $k=1$ to $k=100$.
>     * The improvement provided by AdaCUR over the corresponding AnnCUR baseline tends to be larger at smaller cost budgets and the marginal improvement tends to diminish for larger cost values and for larger values of $k$.
>
>
> **Why is AnnCUR baseline even worse than TF-IDF in Figures 14 and 15?**
>   * Performance of AnnCUR baselines (Yadav et al. 2022) depends on number of anchor queries used for indexing, number of anchor items chosen at test-time and method for choosing anchor items.
>   * Generally performance of CUR-based methods improve with the number of anchor queries and when using 500 or more queries for indexing the items ( see purple bars in Fig. 14b, 14c, 15b, 15c), $\text{AnnCUR}_\text{TF-IDF}$ which uses TF-IDF to sample anchor items, outperforms TF-IDF.
>   * Vanilla variant of AnnCUR baseline (which samples anchor items uniformly at random) sometimes performs worse than TF-IDF baseline under some configurations such as for small values of inference cost budget for the following reason:
>     * Recall that AnnCUR has to split the cost budget between choosing the anchor items and re-ranking items retrieved based on approximate scores.
>     * For instance, at cost=50, AnnCUR uses around 20 to 30 CE calls to score anchor items chosen uniformly at random, and uses the remaining budget of at most 30 CE calls for re-ranking items retrieving based on approximated cross-encoder scores.
>     * Such a small number of anchors chosen at random result in poor approximation of cross-encoder and consequently poor performance when re-ranking up to 30 items retrieved with the approximate scores.
>     * In comparison, TF-IDF based retrieve-and-rerank baselines get to use the entire budget to re-rank 50 items and may perform marginally better than the AnnCUR baseline.
>     * However, note that AnnCUR baseline matches or outperforms TFIDF at larger cost budgets and larger values of $k$

---

### Official Review · Reviewer_6YK8 · 2023-08-05

**Soundness:** 2

**Excitement:**

3: Ambivalent: It has merits (e.g., it reports state-of-the-art results, the idea is nice), but there are key weaknesses (e.g., it describes incremental work), and it can significantly benefit from another round of revision. However, I won't object to accepting it if my co-reviewers champion it.

**Paper Topic And Main Contributions:**

The authors propose ADACUR, a method that effectively minimizes the approximation error for the practically important top-k neighbors by iteratively performing k-NN search using the anchors.

**Questions For The Authors:**

Since the proposed method needs serval rounds, can you provide experiment metrics in terms of average inference time in addition to inference CE calls?

**Reasons To Accept:**

The anchor method is neat for a short paper.

**Reasons To Reject:**

The novelty is at boderline.

**Reproducibility:**

3: Could reproduce the results with some difficulty. The settings of parameters are underspecified or subjectively determined; the training/evaluation data are not widely available.

**Reviewer Confidence:**

4: Quite sure. I tried to check the important points carefully. It's unlikely, though conceivable, that I missed something that should affect my ratings.

---

> ### Author Rebuttal · Authors · 2023-08-28
>
> We would like to thank the reviewer for their time and consideration!
>
> * We observe recall-vs-inference cost trends _similar_ to those currently reported in the paper even when the inference cost is measured as the total inference latency instead of total cross-encoder (CE) calls.
>   * This is because inference latency for AdaCUR (at 5-10 rounds) and other retrieve-and-rerank baselines largely depends on the number of cross-encoder calls made at test-time to score query-item pairs.
>   * Figure 4 shows inference latency (in seconds) of our proposed approach AdaCUR at different CE call budgets.
>   * Thin blue bars in Figure 4 show per-query inference latency (in seconds) on primary y-axis. We show breakdown on inference latency into different components such cross-encoder (CE) score computation and overhead incurred by our proposed approach AdaCUR on secondary y-axis.
>   * Figure 4 shows that AdaCUR has very small overhead when using 5-10 rounds for inference with the time taken to compute cross-encoder scores dominating the overall inference latency.
>
> We would be happy to further discuss and clarify any concerns that the reviewer may have regarding the soundness of our paper.

---

### Official Review · Reviewer_zs87 · 2023-08-06

**Soundness:** 4

**Excitement:**

3: Ambivalent: It has merits (e.g., it reports state-of-the-art results, the idea is nice), but there are key weaknesses (e.g., it describes incremental work), and it can significantly benefit from another round of revision. However, I won't object to accepting it if my co-reviewers champion it.

**Paper Topic And Main Contributions:**

Authors propose to improve upon previously proposed method of annCUR. This is achieved by improving the set of anchor items over multiple rounds of retrieval, rather than using a one round of randomly sampled items. Experiments on zero-shot entity linking dataset (ZeShEl) and BeIR against annCUR as well as other strong dual encoder-based baselines demonstrate that the proposed method more effectively utilize the computation budget for cross-encoder calls than baselines do.

**Reasons To Accept:**

The proposed method is well-motivated by conceptual argument (multiple rounds of adaptive anchors should be better than one round of anchors) as well as empirical analysis of the problem (Figure 1). This improves the understanding on annCUR and also justifies the design of the proposed approach.

Authors compare the proposed method against strong baselines, in particular dual encoders distilled from cross-encoders. This helps readers to understand the practical utility of the proposed method against well-established and strong alternative methods.

**Reasons To Reject:**

I find the analysis of the latency to be a bit confusing or misleading. Authors argue that the main bottleneck is CE computation, and thus using a multiple rounds of retrieval in adaCUR doesn't impact latency much. However, CE calls shall be parallelized, if needed across multiple devices/hosts. Hence, given sufficient compute budget, 100 CE calls shall be run in the latency close to 1 CE calls, whereas 100 rounds of adaCUR will require the latency of 100 one-by-one calls. Therefore, the latency analysis in Figure 4 seems to be conducted in a very unrealistic assumption that only one CE call shall be made at a time.

---

Authors argue that batch parallelization (of batch size 50) only provides 2x speedup. I am a bit surprised by this, but increasing the score based on the assumption the baseline implementation is sufficiently optimized.

**Reproducibility:**

4: Could mostly reproduce the results, but there may be some variation because of sample variance or minor variations in their interpretation of the protocol or method.

**Reviewer Confidence:**

4: Quite sure. I tried to check the important points carefully. It's unlikely, though conceivable, that I missed something that should affect my ratings.

---

> ### Author Rebuttal · Authors · 2023-08-28
>
> Thank you for your thoughtful comments and time reviewing our work.
>
> **Analysis of the latency - Clarifications**
> * Figure 4 reports the total time of each round (thin blue bar) on primary y-axis and the fraction of that time spent in cross-encoder (CE) calls and the other parts of AdaCUR (matmul, pseudo-inverse) on secondary y-axis
> * The reported numbers already batch/parallelize the CE calls as the reviewer suggests. They are not scored one-by-one.
> * The batch size used is 50 (pairs of query-item to score with the CE).
> * Increasing batch size beyond 50 did not yield any improvement in amortized time per score computation on Nvidia 2080ti GPU.
> * Batch Parallel takes: ~310ms to score a batch of size 50 resulting in an amortized time of 6-7ms per query-item pair. Scoring each query-item pair sequentially (with batch size=1) would have taken ~13ms per query-item pair, and effectively ~650ms to score 50 query-item pairs.
>
>
> *AdaCUR compared to Retrieve & Rerank - Single GPU Machine*
>
> Consider cross-encoder (CE) call budget = 500
>
> * Time taken by retrieve-and-rerank based approaches to re-rank 500 retrieved items using cross-encoder = 3100ms (in 10 batches of size 50 each)
>
> * AdaCUR will perform retrieval over 5 rounds, requiring computation of 100 CE scores per round.
> * Time taken for 100 CE calls per round =  620ms (in 2 batches of size 50 each)
> * Total time taken for 500 CE calls over 5 rounds = 3100ms
>
> * Total overhead of AdaCUR over 5 rounds = 42ms
> * Thus, total time taken by AdaCUR = 3100 +  42 = 3142ms
>
> * AdaCUR incurs less than 2% overhead over retrieve-and-rerank based approaches.
>
> *Analysis of the latency - AdaCUR compared to Retrieve & Rerank - Multiple GPUs*
>
> Using multiple machines, say `M`, can further bring down per-query latency by a factor of `M` for retrieve-and-rerank approaches.
> Similar improvement can be observed for AdaCUR when number of CE calls per round > `batch_size*M` i.e. for the same query,  multiple batches of items can be scored in parallel.
>
> In case number of CE calls per round < `batch_size*M`, then some fraction of the machines will be under-utilized. In such cases, we can increase the overall throughput by performing k-NN search multiple queries in parallel.
>
>
> **Impact of model size on inference latency**
>
> We used 12-layer transformer-based models in our experiments which took an amortized time of around 6-7ms per query-item pair when the scores are computed in batches of size=50.
>
> For larger, billion-parameter sized models, the cost of cross-encoder calls increases further and thus the overhead of our proposed adaptive sampling would be further dwarfed in comparison to the cost and latency of each cross-encoder call.
>
> We will add this clarification in the paper.

---

### Meta-Review · Area_Chair_ZRA6 · 2023-09-28

**Recommendation:** 3

**Metareview:**

This paper presents ADACUR to improve upon previously proposed method of annCUR  by improving the set of anchor items over multiple rounds of retrieval. The motivation of this paper is mostly acknowledged by the reviewers, however, there are major concerns on the novelty, comparison or discussion with recent literature, and the presentation (many parts are unclear and there are numerous grammars). Significant improvement over the quality of this paper is required.

---

### Decision · Program_Chairs · 2023-10-07

**Decision:**

Accept-Findings

**Comment:**

This paper presents ADACUR to improve upon previously proposed method of annCUR  by improving the set of anchor items over multiple rounds of retrieval. The motivation of this paper is mostly acknowledged by the reviewers, however, there are major concerns on the novelty, comparison or discussion with recent literature, and the presentation (many parts are unclear and there are numerous grammars). Significant improvement over the quality of this paper is required.